# Objective and efficient inference for couplings in neuronal networks

**Yu Terada**[1,2]**, Tomoyuki Obuchi**[2]**, Takuya Isomura**[1]**, Yoshiyuki Kabashima**[2]
[1]Laboratory for Neural Computation and Adaptation,
RIKEN Center for Brain Science,
2-1 Hirosawa, Wako, Saitama 351-0198, Japan
[2]Department of Mathematical and Computer Science
Tokyo Institute of Technology
Tokyo 152-8550, Japan
`yu.terada@riken.jp, obuchi@c.titech.ac.jp,`
`takuya.isomura@riken.jp, kaba@c.titech.ac.jp`

## Abstract

Inferring directional couplings from the spike data of networks is desired in various scientific fields such as neuroscience. Here, we apply a recently proposed objective procedure to the spike data obtained from the Hodgkin–Huxley type models and *in vitro* neuronal networks cultured in a circular structure. As a result, we succeed in reconstructing synaptic connections accurately from the evoked activity as well as the spontaneous one. To obtain the results, we invent an analytic formula approximately implementing a method of screening relevant couplings. This significantly reduces the computational cost of the screening method employed in the proposed objective procedure, making it possible to treat large-size systems as in this study.

## 1 Introduction

Recent advances in experimental techniques make it possible to simultaneously record the activity of multiple units. In neuroscience, multi-electrodes and optical imaging techniques capture large-scale behaviors of neuronal networks, which facilitate a deeper understanding of the information processing mechanism of nervous systems beyond the single neuron level [1-6]. This preferable situation, however, involves technical issues in dealing with such datasets because they usually consist of a large amount of high-dimensional data which are difficult to be handled by naive usages of conventional statistical methods.

A statistical-physics-based approach for tackling these issues was presented using the Ising model [7]. Although the justification to use the Ising model for analyzing neuronal systems is not completely clear [8,9,10], its performance was empirically demonstrated [7], which triggered further applications [11-22]. An advantage of using the Ising model is that several analytical techniques for inverse problems are available [23-29], which allows us to infer couplings between neurons with a feasible computational cost. Another advantage is that it is straightforward to introduce variants of the model. Beyond the conventional data analysis, an important variant is the kinetic Ising model, which is more suitable to take into account the correlations in time, since this extended model removes the symmetric-coupling constraint of the Ising model. A useful mean-field (MF) inverse formula for the kinetic Ising model has been presented in [25,26].

Two problems arise when treating neuronal systems' data in the framework of the Ising models. The first problem is how to determine an appropriate size of time bins when discretizing original signals in time; the appropriate size differs from the intrinsic time-scale of the original neuronal sys-

tems because the Ising models are regarded as a coarse-grained description of the original systems. Hence, the way of the transformation to the models of this type is nontrivial. The second problem is extracting relevant couplings from the solution of the inverse problem; unavoidable noises in experimental data contaminate the inferred couplings, and hence, we need to screen the relevant ones among them.

In a previous study [30], an information-theoretic method and a computational-statistical technique were proposed for resolving the aforementioned first and second problems, respectively. Those methods were validated in two cases: in a numerical simulation based on the Izhikevich models and in analyzing *in vitro* neuronal networks. The result is surprisingly good: their synaptic connections are reconstructed with fairly high accuracy. This finding motivates us to further examine the methods proposed in [30].

Based on this motivation, this study applies these methods to the data from the Hodgkin–Huxley model, which describes the firing dynamics of a biological neuron more accurately than the Izhikevich model. Further, we examine the situation where responses of neuronal networks are evoked by external stimuli. We implement this situation both in the Hodgkin–Huxley model and in a cultured neuronal network of a previously described design [31], and test the methods in both the cases. Besides, based on the previously described MF formula of [25,26], we derive an efficient formula implementing the previous method of screening relevant couplings within a significantly smaller computational cost. In practice, the naive implementation of the screening method is computationally expensive, and can be a bottleneck when applied to large-scale networks. Hence, we exploit the simplicity of the model, and use the advanced statistical processing with reasonable time in this work. Below, we address those three points by employing the simple kinetic Ising model, to efficiently infer synaptic couplings in neuronal networks.

## 2 Inference procedure

The kinetic Ising model consists of $N$ units, $\{s_i\}_{i=1}^N$, and each unit takes bipolar values as $s_i(t) = \pm 1$. Its dynamics is governed by the so-called Glauber dynamics:

$$P\left(\mathbf{s}(t+1)|\mathbf{s}(t); \{J_{ij}, \theta_i(t)\}\right) = \prod_{i=1}^N \frac{\exp\left[s_i(t+1)H_i(t; \{J_{ij}, \theta_i(t)\})\right]}{\exp\left[H_i(t; \{J_{ij}, \theta_i(t)\})\right] + \exp\left[-H_i(t; \{J_{ij}, \theta_i(t)\})\right]}, \quad (1)$$

where $H_i(t)$ is the effective field, defined as $H_i(t) = \theta_i(t) + \sum_{j=1}^N J_{ij}s_j(t)$, $\theta_i(t)$ is the external force, and $J_{ij}$ is the coupling strength from $j$ to $i$. This model also corresponds to a generalized McCulloch–Pitts model in theoretical neuroscience and logistic regression in statistics. When applying this to spike train data, we regard the state $s_i(t) = 1$ (-1) as the firing (non-firing) state. The inference framework we adopt here is the standard maximum-likelihood (ML) framework. We repeat $R$ experiments and denote a firing pattern $\{s_{ir}^*(t)\}_{i=1}^N$ for $t = 1, 2, \cdots, M$ in an experiment $r(= 1, 2, \cdots, R)$. The ML framework requires us to solve the following maximization problem on the variable set $\{J_{ij}, \theta_i(t)\}$:

$$\{\hat{J}_{ij}, \hat{\theta}_i(t)\} = \underset{\{J_{ij}, \theta_i(t)\}}{\arg\max} \left\{ \frac{1}{R} \sum_{r=1}^R \sum_{t=1}^M \log P\left(\mathbf{s}_r^*(t+1)|\mathbf{s}_r^*(t); \{J_{ij}, \theta_i(t)\}\right) \right\}. \quad (2)$$

This cost function is concave with respect to $\{J_{ij}, \theta_i(t)\}$, and hence, a number of efficient solvers are available [32]. However, we do not directly maximize eq. (2) in this study but instead we employ the MF formula proposed previously [25,26]. The MF formula is reasonable in terms of the computational cost and sufficiently accurate when the dataset size $R$ is large. Moreover, the availability of an analytic formula enables us to construct an effective approximation to reduce the computational cost in the post-processing step, as shown in Sec. 2.3.

Unfortunately, in many experimental settings, it is not easy to conduct a sufficient number of independent experiments [33,34], as in the case of Sec. 4. Hence, below we assume the stationarity of any statistics, and ignore the time dependence of $\boldsymbol{\theta}(t)$. This allows us to identify the average over time as the ensemble average, which significantly improves statistics. We admit this assumption is not always valid, particularly in the case where time-dependent external forces are present, although we treat such cases in Sec. 3.2 and Sec. 4.2. Despite this limitation, we still stress that the present approach can extract synaptic connections among neurons accurately, although the existence of the

time-dependent inputs may decrease its performance. Possible directions to overcome this limitation are discussed in Sec. 5.

## 2.1 Pre-processing: Discretization of time and binarization of state

In the pre-processing step, we have to decide the duration of the interval that should be used to transform the real time to the unit time $\Delta\tau$ in the Ising scheme. We term $\Delta\tau$ the bin size. Once the bin size is determined, the whole real time interval $[0, \mathcal{T}]$ is divided into the set of time bins that are labelled as $\{t\}_{t=1}^{M=\mathcal{T}/\Delta\tau}$. Given this set of the time bins, we binarize the neuron states: if there is no spike train of the neuron $i$ in the time bin with a label $t$, then $s_i^*(t) = -1$; otherwise $s_i^*(t) = 1$. This is the whole pre-processing step we adopt, and is a commonly used approach [7].

Determination of the bin size $\Delta\tau$ can be a crucial issue: different values of $\Delta\tau$ may lead to different results. To determine it in an objective way, we employ an information-theory-based method proposed previously [30]. Following this method, we determine the bin size as

$$\Delta\tau_{\mathrm{opt}} = \arg\max_{\Delta\tau} \left\{ \left( \frac{\mathcal{T}}{\Delta\tau} - 1 \right) \sum_{i \neq j} \hat{I}_{\Delta\tau} \left( s_i(t+1); s_j(t) \right) \right\}, \tag{3}$$

where $I_{\Delta\tau} \left( s_i(t+1); s_j(t) \right)$ denotes the mutual information between $s_i(t+1)$ and $s_j(t)$ in the coarse-grained series with $\Delta\tau$, and $\hat{I}_{\Delta\tau} \left( s_i(t+1); s_j(t) \right)$ is its plug-in estimator. The explicit formula is

$$\hat{I}_{\Delta\tau} \left( s_i(t+1); s_j(t) \right) = \sum_{(\alpha,\beta)\in\{+,-\}^2} r_{\alpha\beta}(i, t+1; j, t) \log \frac{r_{\alpha\beta}(i, t+1; j, t)}{r_\alpha(i, t+1)r_\beta(j, t)}, \tag{4}$$

where $r_{++}(i, t+1; j, t)$ denotes the realized ratio of the pattern $(s_i(t+1), s_j(t)) = (+1, +1)$, $r_{++}(i, t+1; j, t) \equiv (1/(M-1))\#\{(s_i(t+1), s_j(t)) = (+1, +1)\}$, and the other double-subscript quantities $\{r_{+-}, r_{-+}, r_{--}\}$ are defined similarly. Single-subscript quantities are also the realized ratios of the corresponding state, for example, $r_+(j, t) \equiv (1/M)\#\{s_j(t) = +1\}$.

The meaning of eq. (3) is clear: the formula inside the brace brackets of the right-hand side, hereafter termed gross mutual information, is merely the likelihood of a (null) hypothesis that $s_i(t+1)$ and $s_j(t)$ are firing without any correlation. The optimal value $\Delta\tau_{\mathrm{opt}}$ is chosen to reject this hypothesis most strongly. This can also be regarded as a generalization of the chi-square test.

## 2.2 Inference algorithm: The MF formula

The previously derived MF formula [25,26] is given by

$$\hat{J}^{\mathrm{MF}} = A^{-1}DC^{-1}, \tag{5}$$

where

$$\begin{cases} \mu_i(t) = \langle s_i(t) \rangle, \\ A_{ij}(t) = \left( 1 - \mu_i^2(t) \right) \delta_{ij}, \\ C_{ij}(t) = \langle s_i(t)s_j(t) \rangle - \mu_i(t)\mu_j(t), \\ D_{ij}(t) = \langle s_i(t+1)s_j(t) \rangle - \mu_i(t+1)\mu_j(t). \end{cases} \tag{6}$$

Note that the estimate $\hat{J}^{\mathrm{MF}}$ seemingly depends on time, but it is known that the time dependence is very weak and ignorable. Once given $\hat{J}^{\mathrm{MF}}$, the MF estimate of the external field is given as

$$\hat{\theta}_i^{\mathrm{MF}}(t) = \tanh^{-1} \left( \mu_i(t+1) \right) - \sum_j \hat{J}_{ij}^{\mathrm{MF}} \mu_j(t), \tag{7}$$

although we focus on the couplings between neurons and do not estimate the external force in this study. The literal meaning of the brackets is the ensemble average corresponding to $(1/R)\sum_{r=1}^R$ in eq. (2), but here we identify it as the average over time. Here, we use the time-averaged statistics of $\{\boldsymbol{\mu}, C, D, \boldsymbol{\theta}\}$, as declared above.

## 2.3 Post-processing: Screening relevant couplings and its fast approximation

The basic idea of our screening method is to compare the coupling estimated from the original data with the one estimated from randomized data in which the time series of firing patterns of each neuron is randomly independently permuted. We do not explain the detailed procedures here because similar methods have been described previously [7,30]. Instead, here we state the essential point of the method and derive an approximate formula implementing the screening method in a computationally efficient manner.

The key of the method is to compute the probability distribution of $\hat{J}_{ij}$, $P(\hat{J}_{ij})$, when applying our inference algorithm to the randomized data. Once we obtain the probability distribution, we can judge how unlikely our original estimate is as compared to the estimates from the randomized data. If the original estimate is sufficiently unlikely, we accept it as a relevant coupling; otherwise, we reject it.

Evaluation of the above probability distribution is not easy in general, and hence, it is common to have recourse to numerical sampling, which can be a computational burden. Here, we avoid this problem by computing it in an analytical manner under a reasonable approximation.

For the randomized data, we may assume that two neurons $s_i$ and $s_j$ fire independently with fixed means $\mu_i$ and $\mu_j$, respectively. Under this assumption, by the central limit theorem, each diagonal component of $C$ converges to $C_{ii} = 1 - \mu_i^2 = A_{ii}$, while its non-diagonal component becomes a zero-mean Gaussian variable whose variance is proportional to $1/(M-1)$, and is thus, small. All the components of $D$ behave similarly to the non-diagonal ones of $C$. This consideration leads to the expression

$$\hat{J}_{ij}^{\mathrm{ran}} = \sum_k (A^{-1})_{ii} D_{ik} (C^{-1})_{kj} \approx (A^{-1})_{ii} D_{ij} (A^{-1})_{jj} = \frac{1}{(1-\mu_i^2)(1-\mu_j^2)} D_{ij}. \tag{8}$$

By the independence between $s_i$ and $s_j$, the variance of $D_{ij}$ becomes $(1-\mu_i^2)(1-\mu_j^2)/(M-1)$. Hence the probability $P\left(|\hat{J}_{ij}^{\mathrm{ran}}| \geq \Phi_{\mathrm{th}}\right)$ is obtained as

$$P\left(|\hat{J}_{ij}^{\mathrm{ran}}| \geq \Phi_{\mathrm{th}}\right) \approx 1 - \mathrm{erf}\left(\Phi_{\mathrm{th}}\sqrt{\frac{(1-\mu_i^2)(1-\mu_j^2)(M-1)}{2}}\right), \tag{9}$$

where $\mathrm{erf}(x)$ is the error function defined as

$$\mathrm{erf}(x) \equiv \frac{2}{\sqrt{\pi}} \int_0^x dy\, e^{-y^2}. \tag{10}$$

Inserting the absolute value of the original estimate of $\hat{J}_{ij}$ in $\Phi_{\mathrm{th}}$, we obtain its likelihood, and can judge whether it should be accepted. Below, we set the significance level $p_{\mathrm{th}}$ associated with $(\Phi_{\mathrm{th}})_{ij}$ as

$$(\Phi_{\mathrm{th}})_{ij} = \sqrt{\frac{2}{(1-\mu_i^2)(1-\mu_j^2)(M-1)}}\, \mathrm{erf}^{-1}\left(1 - p_{\mathrm{th}}\right) \tag{11}$$

and accept only $\hat{J}_{ij}$ such that $|\hat{J}_{ij}| > (\Phi_{\mathrm{th}})_{ij}$.

## 3 Hodgkin–Huxley networks

We first evaluate the accuracy of our methods using synthetic systems consisting of the Hodgkin–Huxley neurons. The dynamics of the neurons are given by

$$C\frac{dV_i}{d\tau} = -\bar{g}_{\mathrm{K}} n_i^4 \left(V_i - E_{\mathrm{K}}\right) - \bar{g}_{\mathrm{Na}} m_i^3 h_i \left(V_i - E_{\mathrm{Na}}\right) - \bar{g}_{\mathrm{L}} \left(V_i - E_{\mathrm{L}}\right) + I_i^{\mathrm{ex}}, \tag{12}$$

$$\frac{dn_i}{d\tau} = \alpha_n\left(V_i\right)\left(1 - n_i\right) - \beta_n\left(V_i\right) n_i, \tag{13}$$

$$\frac{dm_i}{d\tau} = \alpha_m\left(V_i\right)\left(1 - m_i\right) - \beta_m\left(V_i\right) m_i, \tag{14}$$

$$\frac{dh_i}{d\tau} = \alpha_h\left(V_i\right)\left(1 - h_i\right) - \beta_h\left(V_i\right) h_i, \tag{15}$$

where $V_i$ is the membrane potential of $i$th neuron, $n_i$ is the activation variable that represents the ratio of the open channels for K$^+$ ion, and $m_i$ and $h_i$ are the activation and inactivation variables for Na$^+$ ion, respectively. All parameters, except the external input term $I_i^{\text{ex}}$, are set as described in [35]. The input forces are given by

$$I_i^{\text{ex}} = c_i(\tau) + \sum_{j=1}^{N} K_{ij} V_j \Theta\left(V_j - V_{\text{th}}\right) + a \sum_k \delta\left(\tau - \tau_i^k\right), \tag{16}$$

where $c_i(t)$ represents the environmental noise with a Poisson process, the second term represents the couplings with the threshold voltage $V_{\text{th}} = 30\,\text{mV}$ and the Heaviside step function $\Theta(\cdot)$, and the last term denotes the impulse stimulations with the delta function. Here, we consider no-delay simple couplings, which we term the synaptic connections, and aim to reconstruct their structure with the excitatory/inhibitory signs using our methods. We use $N = 100$ neuron networks, where the 90 neurons are excitatory and have positive outgoing couplings while the others are inhibitory. The rate and strength of the Poisson process are set as $\lambda = 180\,\text{Hz}$ and $b = 2\,\text{mV}$, respectively, for all neurons. We generate their time series, integrating (12)-(15) by the Euler method with $d\tau = 0.01\,\text{ms}$, where we suppose a neuron is firing when its voltage exceeds $V_{\text{th}}$, and use the spike train data with the whole period $\mathcal{T} = 10^6\,\text{ms}$ for our inference.

## 3.1 Spontaneous activity case

At first, we consider a system on a chain network in which each neuron has three synaptic connections to adjoint neurons in one direction. The connection strength $K_{ij}$ is drawn from the uniform distributions in $[0.015, 0.03]$ for the excitatory and in $[-0.06, -0.03]$ for the inhibitory neurons, respectively. Here, we set $a = 0\,\text{mV}$ to study the spontaneous activity. An example of the spike trains generated during 3 seconds is shown in Fig. 1 (a), where the spike times and corresponding neuronal indices are plotted. Subsequently, using the whole spike train data, we calculate the gross mutual information for different $\Delta\tau$, and the result is indicated by the red curve in Fig. 1 (b). The curve has the unimodal feature, which implies the existence of the optimal time bin size of approximately $\Delta\tau = 3\,\text{ms}$, although the original system does not have the delay. We suppose that inputs must accumulate sufficiently to generate a spike, which costs some time scale, and this is a possible reason for the emergence of the nontrivial time-scale. To validate our approximation (8), we randomize the coarse-grained series with $\Delta\tau = 3\,\text{ms}$ in the time direction independently, rescale $\hat{J}_{ij}^{\text{ran}}$ by multiplying $\sqrt{(1 - \mu_i^2)(1 - \mu_j^2)(M - 1)}$, and compare the results of 1000 randomized data with the standard Gauss distribution in Fig. 1 (c), which shows their good correspondence. Using $\Delta\tau = 3\,\text{ms}$ to make the spike trains coarse-grained, we apply the inverse formula to the series and screen relevant couplings with $p_{\text{th}} = 10^{-3}$, which leads to the estimated coupling matrix shown in Fig. 1 (e), while the one used to generate the data is shown in Fig. 1 (d). The asymmetric network structure is recovered sufficiently with the discrimination of the signs of the couplings. The conditional ratios of the correctness are shown in Fig. 1 (f), where the inference results obtained with different values of $\Delta\tau$ are also shown. This demonstrates the fairly accurate reconstruction result obtained using our inference procedure. We also show the receiver operating characteristic (ROC) curves obtained by gradually changing the value $p_{\text{th}}$ in Fig. 1 (g), with the different values of $\Delta\tau$. We conclude that using non-optimal time bins drastically decreases the accuracy of the inference results.

To illustrate the robustness of the optimality of the time bin, in Fig. 1 (i) we plot the means and standard deviations of the gross mutual information through the 10 different simulations, showing that the variance is small enough and the result is well robust.

To consider a more general situation, we also employ a Hodgkin–Huxley system on a random network. The directional synaptic connection between every pair of neurons is generated with the probability 0.1, and the excitatory and inhibitory couplings are drawn from the uniform distributions within $[0.01, 0.02]$ and $[-0.04, -0.02]$, respectively. The corresponding inference results for its spontaneous activity are shown by green curves in Figs. 1 (b) and (f). The ROC curves for the three different three values of $\Delta\tau$ are also shown in (h). We confirm that the inference is sufficiently effective in the random-network system as well as in the chain system.

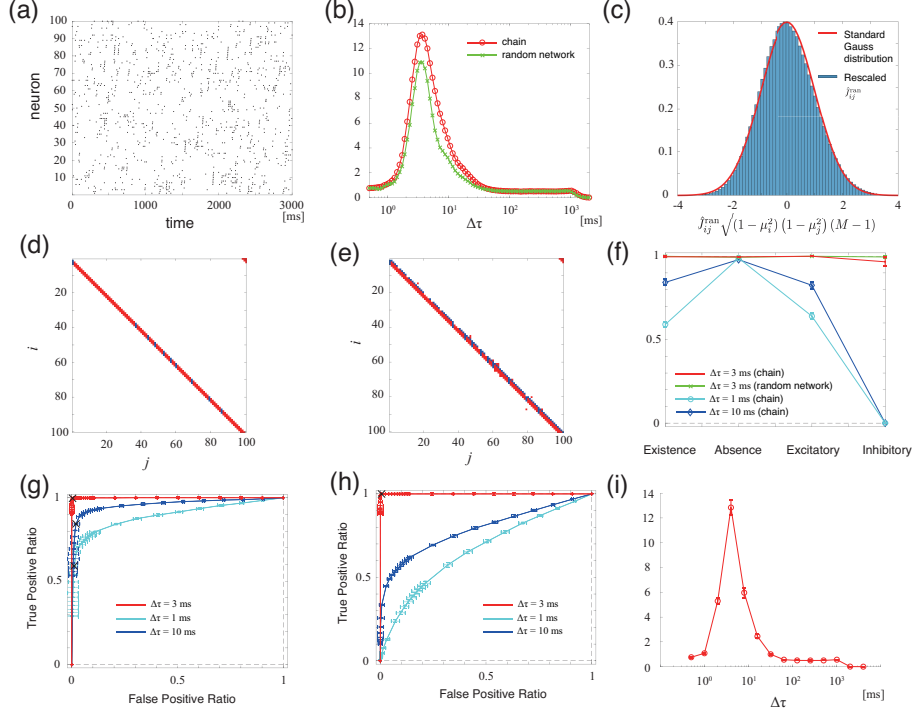

Figure 1: Application of the proposed approach to the Hodgkin–Huxley models. (a) Spontaneous spike trains during 3 seconds. (b) Gross mutual information v.s. time bin size $\Delta\tau$. The red curve shows the chain network while the green curve shows the random network. (c) Histogram of rescaled $\hat{J}_{ij}^{\mathrm{ran}}$ obtained by randomizing the original series, and the standard Gauss distribution. (d) An example of the chain networks that we used, where the red and blue elements indicate the excitatory and inhibitory couplings, respectively. (e) Corresponding inferred coupling network with $\Delta\tau = 3\,\mathrm{ms}$. (f) Conditional correctness ratios for the existence, absence, excitatory coupling, and inhibitory coupling, where the standard deviations of 10 different simulations are shown with the error bars. (g,h) Receiver operating characteristic curves for different coarse-grained series in the systems (g) on the chain and (h) on the random network, where the error bars indicate the standard deviations of 10 different simulations. The marked points indicate $p_{\mathrm{th}} = 10^{-3}$ used in (e) and (f). (i) The mean and standard deviation of the gross mutual information for 10 independent simulations of the chain systems. The result is shown to be robust.

## 3.2 Evoked activity case

We next investigate performance in systems where responses are evoked by impulse stimuli. The model parameters, except for $a$, are the same as those in the chain model in Sec. 3.1. The strength of the external force is set as $a = 5.3\,\mathrm{mV}$, and the stimulations are injected to all neurons with interval $1\,\mathrm{s}$. In Fig. 2 (a) we show the spike trains, where we observe that most of the neurons fire at the injection times $\tau = 0.5, 1.5, 2.5\,\mathrm{s}$. The gross mutual information against $\Delta\tau$ is shown in Fig. 2 (b). Although the curve feature is modified due to the existence of the impulse inputs, we observe that its peak is located at a similar value of $\Delta\tau$. Therefore, we use the same value $\Delta\tau = 3\,\mathrm{ms}$. Applying our inference procedure with $\Delta\tau = 3\,\mathrm{ms}$ and $p_{\mathrm{th}} = 10^{-3}$, we obtain the inferred couplings which are shown in Fig. 2 (c), where the original network is in Fig. 1 (d). On comparing Fig. 2 (c) with Fig. 1 (e), while the inference detects the existence of the synaptic connections, we observe more false couplings in the evoked case. The conditional ratios in Fig. 2 (d) indicate that the existence of the external inputs may increase the false positive rate with the same $p_{\mathrm{th}}$. The ROC curves are shown in Fig. 2 (f).

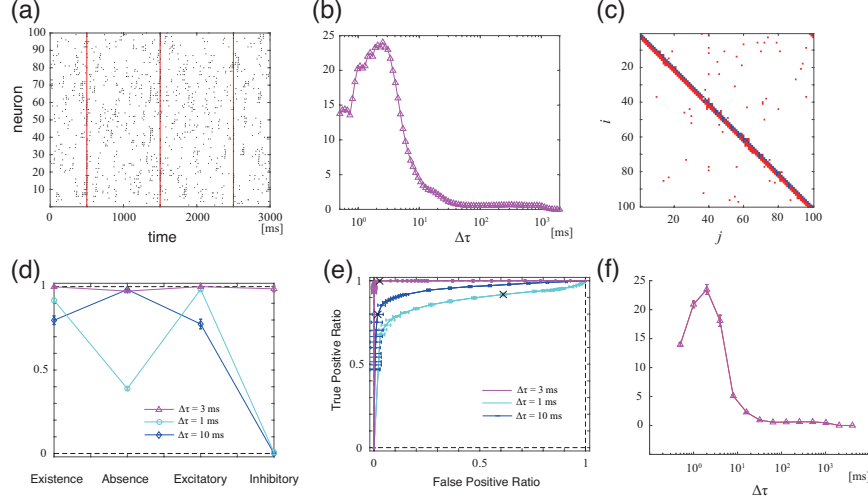

Figure 2: Application of the proposed approach to the evoked activity in the Hodgkin–Huxley models. (a) Evoked spike trains during 3 seconds, where the red line expresses the injection times of the stimuli. (b) Gross mutual information v.s. time bin size. (c) Inferred coupling matrix with the red excitatory and blue inhibitory elements using $\Delta\tau = 3\,\mathrm{ms}$, where the generative network is the one shown in Fig. 1 (b). (d) Conditional correctness ratios. (e) Receiver operating characteristic curves for different coarse-grained series, where the points denoting $p_{\mathrm{th}} = 10^{-3}$ are marked. (f) The mean and standard deviation of the gross mutual information for 10 independent simulations. The result is shown to be robust.

## 4 Cultured neuronal networks

We apply our inference methods to the real neuronal systems introduced in a previous study [31], where rat cortical neurons were cultured in micro wells. The wells had a circular structure, and consequently the synapses of the neurons were likely to form a physically asymmetric chain network, which is similar to the situation in the Hodgkin–Huxley models we used in Sec. 3. The activity of the neurons was recorded by the multi-electrode array with $40\,\mu$s time resolution, and the Efficient Technology of Spike sorting method [36] was used to identify the spike events of individual neurons. We study the spontaneous and evoked activities here.

### 4.1 Spontaneous activity case

We first use the spontaneous activity data recorded during $120\,$s. The spike sorting identified 100 neurons which generated the spikes. The spike raster plot during 3 seconds is displayed in Fig. 3 (a). We calculate the gross mutual information as in case of the Hodgkin–Huxley models, and the obtained optimal bin size is approximately $\Delta\tau = 5\,$ms. We also confirm that the inferred couplings are similar to the results described previously [30], and this supports the validity of our novel approximation method introduced in Sec. 2.3. We show the inferred network in Figs. 3 (b-d) with different values $p_{\mathrm{th}} = 10^{-3}, 10^{-6}, 10^{-9}$, where we locate the nodes denoting the neurons on a circle following the experimental design [31]. A more strict threshold provides us with clear demonstration of the relevant couplings here.

### 4.2 Evoked activity case

We next study an evoked neuronal system, where an electrical pulse stimulation is injected from an electrode after every 3 seconds, and the other experimental settings are similar to those of the spontaneous case. In this case the activity of 149 neurons were identified by the spike sorting. The example of the spike trains is shown in Fig. 4 (a). The gross mutual information is shown in Fig. 4 (b), where we can see the peak around $\Delta\tau = 10\,$ms. Setting $\Delta\tau = 10\,$ms and $p_{\mathrm{th}} = 10^{-3}, 10^{-6}$, we obtain the estimated coupling matrices in Figs. 4 (c,d). In these cases, we can also observe the bold diagonal elements representing the asymmetric chain structure, although with the lower

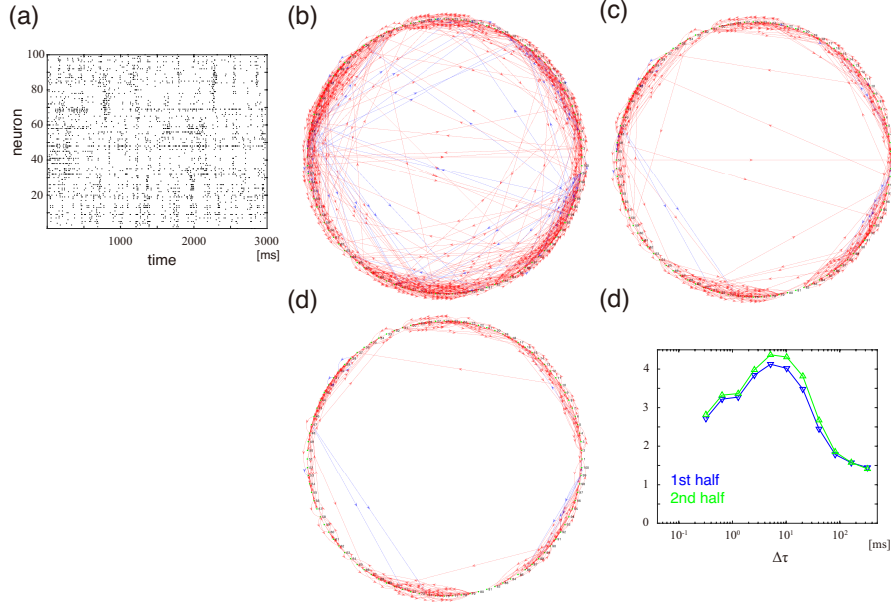

Figure 3: Application of the proposed approach to a cultured-neuronal system. (a) Spike trains during 3 seconds. (b-d) Inferred networks, where the nodes are located on the circle corresponding to the experimental design. The different significant levels used are: (b) $10^{-3}$, (c) $10^{-6}$, and (d) $10^{-9}$. The red and blue directional arrows represent the excitatory and inhibitory couplings, respectively. (e) The gross mutual information for the 1st and 2nd halves of the data. The figure shows the robustness of the result.

significant level some far-diagonal elements emerge due to the existence of the external inputs, which is a situation similar to that in the Hodgkin–Huxley simulation in Sec. 3.2. The inferred network with the strict threshold $p_{\text{th}} = 10^{-9}$ is displayed in Fig. 4 (e), where some long-range couplings are still estimated while physical connections corresponding to them do not exist because of the experimental design.

## 5 Conclusion and discussion

We propose a systematic inference procedure for extracting couplings from point-process data. The contribution of this study is three-fold: (i) invention of an analytic formula to screen relevant couplings in a computationally efficient manner; (ii) examination in the Hodgkin–Huxley model, with and without impulse stimuli; (iii) examination in an evoked cultured neuronal network.

The applications to the synthetic data, with and without the impulse stimuli, demonstrate the fairly accurate reconstructions of synaptic connections by our inference methods. The application to the real data of the spontaneous activity in the cultured neuronal system also highlights the effectiveness of the proposed methods in detecting the synaptic connections.

From the comparison between the analyses of the spontaneous and evoked activities, we found that the inference accuracy becomes degraded by the external stimuli. One of the potential origins is the breaking of our stationary assumption of the statistics $\{\boldsymbol{\mu}, C, D\}$ because of the time-varying external force $\boldsymbol{\theta}$. To overcome this, certain techniques resolving the insufficiency of samples, such as regularization, will be helpful. A promising approach might be the introduction of an $\ell_1$ regularization into eq. (2), which enables us to automatically screen out irrelevant couplings. Comparing it with the present approach based on computational statistics will be an interesting future work.

### Acknowledgments

This work was supported by MEXT KAKENHI Grant Numbers 17H00764 (YT, TO, and YK) and 18K11463 (TO), and RIKEN Center for Brain Science (YT and TI).

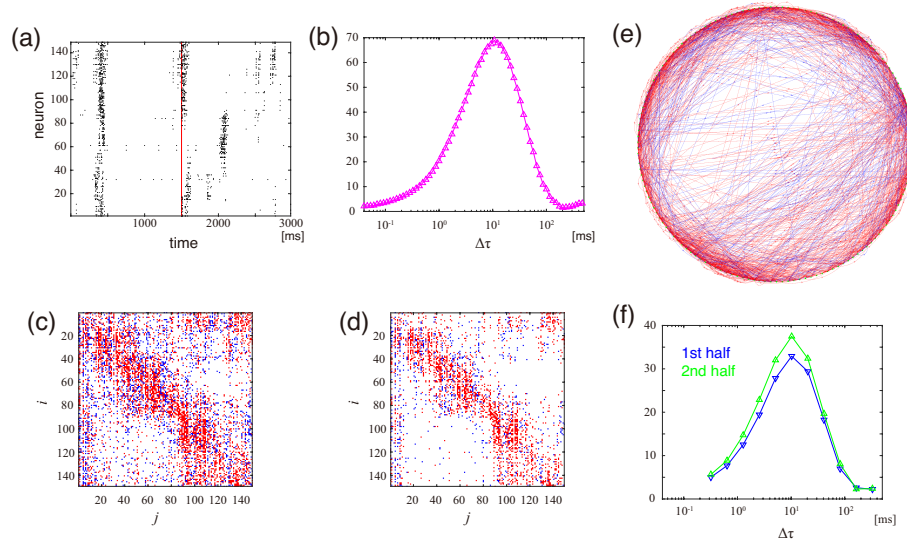

Figure 4: Application of the proposed approach to an evoked cultured-neuronal system. (a) Spike trains during 3 seconds, where the red line indicates the injection time. (b) Gross mutual information v.s. time bin size. (c,d) Inferred coupling matrices for (c) $p_{\mathrm{th}} = 10^{-3}$ and (d) $p_{\mathrm{th}} = 10^{-6}$. (e) Inferred network with $p_{\mathrm{th}} = 10^{-9}$. (f) The gross mutual information for the 1st and 2nd halves of the data. The figure shows the robustness of the result.

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
