[Reviews · NeurIPS 2018]

Reviewer 1



SUMMARY OF THE PAPER Dealing with a relevant research problem -reconstruction of synaptic connections-; The authors put forth a new approximation to estimate neural couplings from spike data with implications for the analysis of functional connectivity. It took me a bit to understand what was the intended goal (aim or hypothesis never stated explicitly and contributions are declared at the end), but as soon as the intention was clear, the rest of the paper reads quite well. In their approximation, the authors make some assumptions (e.g. time invariance) are strict but otherwise acceptable and reasoned, and if I may, I would proceed just like the authors. The simulations on synthetic data are very informative and the results from the cultured cells gives a good idea of some of the practical limitations of the approach. I’m not sure I share the idea that some coactivity threshold should be based on a significance test, but then, it is not that I have a better recommendation or harder facts, so this latter it is only my humble opinion. STRONG POINTS • Good and well explained experiments, both on synthetic data and cultured cells covering both spontaneous and evoked activity • Acceptable and well reasons decisions whenever there has been one • The contributed approximation for analysing neural couplings is an important contribution that (I understand) have low computational burden. WEAK POINTS • Test on single topology • At times, the author seems to favour mathematical decisions over biological dictum (e.g. why should the optimal binning be based on hypothesis testing and not on some physiological guidance?). • Criticism of models such as the Ising models seem more concerned on justifying the authors choice than actually about pointing out a genuine limitation for the model and analysis SUGGESTIONS TO (PERHAPS) IMPROVE THE PAPER Major • Well, mostly the first two afore stated as weak points above, and yet none is a critical impediment for acceptance; testing and reporting on more topologies will make the paper too dense, and my concerns on favouring some mathematical commodities over phenomenological construct are understandable within the specific frame of the proposal. I’m happy leaving the attendance of this two to the authors’ criterium. Minor • Avoid adjectives (e.g. difficult, naïve, etc) and whenever possible rely on objective (preferably quantitative) measures. • Line 230: Any patterns that the authors may appreciate on Fig 4? For instance, is there any links corresponding to neurons “seeing” the spike train at a particular lag? Any preferred anisotropy direction in the transfer of information across neurons worth mentioning? Does the distribution of error/noise follows any known distribution? • If not critical for the authors purposes and writing styles, I would suggest moving the contributions the introduction and perhaps declared explicitly the goal (providing a new approximation affording some specific benefit) from the abstract.

Reviewer 2



This paper improves on previous work on capturing the directional coupling spike behaviour of large neuronal systems. Previous work used synthetic data from the Izhikevich model of neuronal activity behaviour (a simpler model than the well-known Hodgkin-Huxley set of differential equations), the kinetic Ising model and mean field theory for approximate inference. The present paper develops a slightly simplified analytical solution to the equations, which are subsequently applied to the more sophisticated Hodgkin-Huxley neuronal activity model i(the synthetic data part) and real data from cortical neurons of the rat. It appears that although not perfect, the mathematical model performs well when choosing the right parameters. Quality; well-written and relatively easy to follow paper that has a clear goal and outcome. Clarity: clear structure. Content presumes quite some basic knowledge in neuroscience research, so it may not be too clear for people outside of neuroscience. Originality: the main contribution is the analytical formula (I have no idea whether this is original; I can immagine that similar ideas have been stated before), where the rest of the paper follows previously done research, however in a scholarly way. Significance: moderate.

Reviewer 3



This contribution proposes a method to reconstruct the connection in a neural network based on the spike recording. The method infers coupling, only considering relevant connections. The proposed method is evaluated on simulated data, generated by Hodgkin-Huxley neurons and on real data obtained with cultured neuronal networks. The experiments on real data is appreciated and the obtained results are impressive, as the reconstructed networks seems very accurate. Some of the limitations of this work concern the careful selection of the bin size $\delta \tau$, as a costly parameter validation should be conducted to select the bin size correctly. It could be interesting to show the robustness of the proposed method with respect to bin size. Another limitation is the lack of comparison with state of the art methods, the authors mentioned previous work based on Ising models but omit works relying on transfer entropy [1], on spike metrics [2] or on inverse covariance [3]. Also the bibliography could be shortened up: for example, eleven references are provided for applicative work in one time and are not really detailed. The experimental results indicate that evoked activity led by external stimuli degrades the quality of the estimated connection graph. This could be a consequence of the lack of the system to take into account synaptic delays. How non-null delays are process by this model? [1] Ito, S., Hansen, M. E., Heiland, R., Lumsdaine, A., Litke, A. M., & Beggs, J. M. (2011). Extending transfer entropy improves identification of effective connectivity in a spiking cortical network model. PloS one, 6(11), e27431. [2] Kuroda, K., Fujiwara, K., & Ikeguchi, T. (2012, November). Identification of neural network structure from multiple spike sequences. In International Conference on Neural Information Processing (pp. 184-191). Springer, Berlin, Heidelberg. [3] Mohler, G. (2014). Learning convolution filters for inverse covariance estimation of neural network connectivity. In Advances in Neural Information Processing Systems (pp. 891-899). ---- Edit: After rebuttal period and based on the authors reply, my main questions have been addressed and I changed my overall score appreciation accordingly.